# Dimensions of Thermal Inequity: Neighborhood Social Demographics and Urban Heat in the Southwestern U.S.

**DOI:** 10.3390/ijerph18030941

**Published:** 2021-01-22

**Authors:** John Dialesandro, Noli Brazil, Stephen Wheeler, Yaser Abunnasr

**Affiliations:** 1Geography Graduate Group, University of California Davis, One Shields Ave., Davis, CA 95616, USA; nbrazil@ucdavis.edu (N.B.); smwheeler@ucdavis.edu (S.W.); 2Department of Landscape Design and Ecosystem Management, American University of Beirut, Bliss Street, Ras Beirut, POB 11-0236, Riad El Solh, Beirut 1107 2020, Lebanon; ya20plus@gmail.com

**Keywords:** urban heating, urban heat island, climate justice, thermal inequity, environmental justice, climate adaptation

## Abstract

Exposure to heat is a growing public health concern as climate change accelerates worldwide. Different socioeconomic and racial groups often face unequal exposure to heat as well as increased heat-related sickness, mortality, and energy costs. We provide new insight into thermal inequities by analyzing 20 Southwestern U.S. metropolitan regions at the census block group scale for three temperature scenarios (average summer heat, extreme summer heat, and average summer nighttime heat). We first compared average temperatures for top and bottom decile block groups according to demographic variables. Then we used spatial regression models to investigate the extent to which exposure to heat (measured by land surface temperature) varies according to income and race. Large thermal inequities exist within all the regions studied. On average, the poorest 10% of neighborhoods in an urban region were 2.2 °C (4 °F) hotter than the wealthiest 10% on both extreme heat days and average summer days. The difference was as high as 3.3–3.7 °C (6–7 °F) in California metro areas such as Palm Springs and the Inland Empire. A similar pattern held for Latinx neighborhoods. Temperature disparities at night were much smaller (usually ~1 °F). Disparities for Black neighborhoods were also lower, perhaps because Black populations are small in most of these cities. California urban regions show stronger thermal disparities than those in other Southwestern states, perhaps because inexpensive water has led to more extensive vegetation in affluent neighborhoods. Our findings provide new details about urban thermal inequities and reinforce the need for programs to reduce the disproportionate heat experienced by disadvantaged communities.

## 1. Introduction

Urban heat is a public health hazard that is experienced across the globe independent of jurisdiction, development, and climate. Extreme heat events exacerbate the impact of urban heat and are responsible for a high number of weather-related deaths [1] Excess heat has consequences for human health including heat stroke, exhaustion, and amplified respiratory and cardiovascular issues [2]. Extreme heat is expected to become more common and longer lasting as climate change drives temperatures up.

It has long been known that the paved surfaces of urban areas will absorb and retain solar radiation, increasing urban temperatures [3,4]. For this reason, cities in temperate climates will often have higher temperatures than vegetated surrounding countrysides, a phenomenon known as the urban heat island (UHI) effect [5,6]. Dryland cities as well as tropical cities also experience this phenomena [7]. Likewise, particular neighborhoods with high levels of paved surfaces may be hotter than others with more vegetation.

Heating patterns in dryland urban regions are often more complex than for wetter temperate regions. Dryland cities often contain more green vegetation than surrounding desert or grassland, and so may be somewhat cooler than surrounding landscapes during the daytime [8]. At night, this pattern may be reversed, as desert landscapes cool off more quickly than urban ones. However, during both day and night, temperatures in these cities may be high enough to pose serious problems for human communities.

Urban heat can be reduced by lowering the amount of paved surfaces, increasing their albedo, changing built form in ways that produce ground-level shade, and/or increasing the quantity of vegetation, which cools both through transpiration and by producing shade [9].Vegetation, in particular, has been shown to be a strong cooling influence (e.g., [10,11]).

Thermal inequities have been less explored in the academic literature than other urban heat dynamics. They can be seen as an environmental justice problem in which poor and non-White communities suffer disproportionately because of prevailing environmental conditions. The term “climate justice” is increasingly used [12,13]. More specifically, the climate gap or the unequal impact of the climate crisis on persons and communities of color has received attention as research indicates that communities of color will have a higher degree of suffering during heat waves [14].

Literature on thermal inequities is beginning to emerge, although studies are often general or limited to specific case studies. Mitchell and Chakraborty [15,16] used the term to describe the unequal distribution of the urban heat island impacts amongst populations in Florida. Wong et al. [17] found thermal inequities in age, marital status, occupation, education attainment, and income throughout Hong Kong. Furthermore, Harlan et al. [18] found older, low-income, and minority populations in Phoenix to be disproportionately impacted by increases in urban temperatures. In a study of New York, Chicago, and Los Angeles, Mitchell et al. [19] found higher temperatures in census tracts to be associated with Black, Hispanic, disabled, non-English-speaking, and low-income populations. Mitchell and Chakraborty [13] provided an expanded analysis of 20 cities throughout the U.S. and found census tracts with higher socioeconomic status to be associated with greater heat exposure. However, they found an inconsistent association between minority neighborhoods and greater heat. In a study of Phoenix, Jenerette et al. [20] found that increases in median income of $10,000 were associated with a decrease of 0.28 °C in neighborhood temperature. In a study of Santiago, Chile, De La Barrera et al. [21] found low-income neighborhoods to have temperatures 2.5–3.3 °C higher than in high-income districts.

Within Europe, the Index of Multiple Deprivation (IMD) (a 37-parameter metric that includes education, income, disability, etc.) has been found to correlate with higher temperatures. Macintyre et al. [22] found nighttime temperature to range by 3.1 °C across deciles of the IMD index within West Midlands, UK. Low-income and elderly populations also typically have lesser ability to mitigate heat effects by accessing green spaces and cooling centers [23,24].

Greenspaces with their cooling benefits appear more frequently in high-income White neighborhoods in dry urban regions than in low-income or non-White neighborhoods [25]. Jenerette et al. [26] found vegetation to be positively correlated with income in seven Southwestern U.S. metropolitan regions. Greenspaces are often absent or fragmented in low-income neighborhoods [27]. While finding White neighborhoods to have more tree canopy, other researchers determined that Latinx and African American neighborhoods have higher potential for tree canopy [28,29], indicating that the mitigation potential from tree canopy in Hispanic and African American neighborhoods exceeds that in White neighborhoods.

In a study of Phoenix, Harlan et al. [18] found that resident proximity to greenspaces varies widely between wealthy and poor neighborhoods, as well as between White and Latinx neighborhoods. In arid urban regions, water is a costly resource, and the use of water for aesthetic landscaping is far more apparent in wealthy than in poor neighborhoods, which also leads to cooler neighborhoods [30].

Historic redlining policies (preventing people of color from purchasing homes in certain neighborhoods) heavily influence where tree canopy exists in cities across the U.S. In a recent study, Locke et al. [29] found that neighborhoods previously subject to redlining have 37% less tree canopy, leading to warmer temperatures). Another recent study by Hoffman, Shandas, and Pendleton of 108 U.S. urban areas found that 94% of neighborhoods that had been redlined had elevated temperatures compared to the rest of the city [31]).

Urban greening efforts to reduce heat in neighborhoods with vulnerable populations have been described in the literature [32]. Other strategies to achieve the same result include investing in cool pavements, high albedo roofing, and reconfiguring built form to add ground-level shade [33]. Within the state of California, the California Global Warming Solutions Act (AB 32) and SB 535 Disadvantaged Communities Act mandate the consideration of social equity within climate adaptation programs [34].

In this study, we seek to provide a detailed understanding of thermal inequities within urban regions both at high risk from heat (due to very hot summers) and containing many low-income and non-White neighborhoods. We chose to develop a comprehensive sample of the 20 largest metropolitan areas in the Southwestern U.S. states (Figure 1) and used a more fine-grained geographical scale (the census block group) than previous studies. Phoenix, Arizona, is already heavily represented in the literature on UHI studies, but the remaining 19 urban regions are far less studied. To develop a well-rounded understanding of thermal inequities, we analyzed temperature–demographic relationships on average hot days, extreme heat days, and at nighttime.

## 2. Materials and Methods

### 2.1. Data Acquisition

Utilizing Google Earth Engine (GEE), we downloaded Landsat 8 Operational Land Imager (OLI) Tier 1 imagery. Tier 1 is in a scaled digital number format (often described in the literature as a raw scene). Using JavaScript-enabled GEE we processed these data to obtain at-sensor (or top-of-atmosphere (TOA)) radiance and surface reflectance. We then used TOA radiance and surface reflectance data to retrieve Land Surface Temperature LST. The geometric fidelity of the Tier 1 Landsat Collections has a root mean square error (RSME) of less than 12 m [35]. We projected all imagery into WGS 1984 UTM format for the specific zone of each city. Dates were selected for cloud-free periods (which provide the most accurate readings) in the summer season.

We examined three temperature scenarios for analysis. For the first, we chose Landsat scenes coinciding with extreme heat events for each of the 20 metropolitan regions. This was done by selecting the warmest day available between 2013 and 2019 that coincided with a cloud-free day of Landsat 8 OLI imagery capture. For the second scenario, we collected scenes for an average summer day based on the average of available Landsat scenes during the study period. The dual-level analysis allows for comparison of the urban heat island patterns during extreme heat events as well as during conditions close to the median daytime temperature value for the summer season. For the third scenario, we used nighttime thermal imagery for an average summer temperature night close to the median nighttime temperature value for summer imagery available.

We collected demographic data from the 5-year, 2013–2017 ACS (American Community Survey) at the block-group level. Our primary variables of interest were median household income, percent Latinx, percent Black, and percent Asian. We also collected the total population, percent native-born, and percent holding a bachelor degree as control variables. We removed block groups that had missing values on any variables as well as those in areas with airports and detention facilities Fresno, California is provided below in Figure 2 as an example.

### 2.2. Data Processing

We employed a widely used single-channel method from the Landsat Handbook for LST retrieval. In this method, only TOA (top of atmosphere) radiance and NDVI (normalized difference vegetation index) are required. We converted TOA radiance of the thermal infrared band to TOA (or at-sensor) brightness temperature based on the formula (Equation (1)) [36,37]:Tsensor = K1 / ln (K2 / (Lλ + 1))(1)

This single-channel algorithm is where Tsensor is the at-sensor brightness temperature in Kelvin (K) and L is the TOA radiance in W/m^2^ srμm. For Landsat-8 TIRS, K1 is 774.89 W/(m^2^ srμm) and K2 is 1321.08 K for band 10 [38] K1 and K2 are the band-specific thermal conversion constants. For nighttime imagery, we applied a single-channel algorithm as well but used the closest available daytime date to extract the emissivity values. The following equation (Equation (2)) calculates the LST based on the brightness temperature obtain previously using emissivity [39] and λ is the wavelength in meters and α = 1.438 × $10^{-2}$ mK. ε represents the surface emissivity, which differs for various land cover types [40] For ε, water (NDVI < 0) was assigned a value of 0.9925, urban impervious areas and bare soil (0 ≤ NDVI < 0.15) were assigned a value of 0.923, and vegetation (NDVI > 0.727) was assigned a value of 0.986. Otherwise, there was a modeling relationship with the NDVI values through the following equation:(2)$${\mathrm{LST}\;=\;T}_{\mathrm{sensor}}{\;/\;((1\;+\;\lambda \;*\;T}_{\mathrm{sensor}}\;/\;\alpha )\;*\;\ln(\varepsilon ))$$

### 2.3. Thermal Inequity Analysis

In order to establish a baseline of the relationship between surface temperature and neighborhood median household income and racial/ethnic minority composition throughout the 20 Southwest metropolitan regions, we ran an ordinary least squares (OLS) regression model using all block groups in the 20 metropolitan areas. We ran three models using independent variables of median income, percent Latinx, and percent Black for an extreme heat day, an average heat day, and an average warm-season nighttime temperature. In all cases, LST was used as the dependent variable for urban heat. Because classic regression cannot take into account the variation, we accounted for variation within metropolitan areas by using a fixed effects parameter on the regression (see Table A1: Appendix A). In total, nine OLS models were run.
Temp(scenario)~Demographic Variable+Population+factor(metro)
Example: HotDay ~ Income + Population + factor(metro)(3)

We found spatial autocorrelation of LST in 19 out of our 20 metropolitan regions indicated by statistically significant values of Moran’s I [41]. The urban environment has a complex matrix of surface materials and surface geometry and the orientation of both [41,42] This plus local convection, conduction, and incident solar radiation can cause spatial autocorrelation in LST readings [43]. We controlled for this by running spatial error regression models. In total, we constructed 59 spatial error models to analyze how social indicators of neighborhood demographics influenced the UHI effect: 20 models for extreme heat in each metropolitan area as well as 20 models for average heat. We ran only 19 models for nighttime temperatures due to issues accessing nighttime temperatures for the Dallas metropolitan area.

The inferential statistics yielded from our models produce substantial results indicating thermal inequities between demographic groups. However, certain influences from biophysical variables could not be removed. Differences in elevation and proximity to water bodies, for example, may affect neighborhood temperatures. Literature related to socioeconomic inequities often compares population subsets such as the top and bottom deciles [44]. For our study, using these relatively large subsets of census block groups meant that the effect of such biophysical variables would be minimized. This method also helps convey the extent of disparities between the poorest and wealthiest in society and the wide gaps found today among different demographic groups.

We thus performed a comparison of means by partitioning our data into deciles for each metropolitan area based on demographic values (income, percent Latino, etc.). We examined the average temperatures for block groups falling within the upper 10% of income throughout a metropolitan region to the mean of the lower 10% of income block groups, as well as upper and lower 10% of block groups by Latinx and Black populations.

## 3. Results and Discussion

The OLS regression models were run for our three demographic variables (median income, percent Latinx, and percent Black population) against our three temperature scenarios. In each equation, population was also used as a control factor. The R^2^ values were all high, indicating strong correlation (Table A1); however, there was spatial autocorrelation in the dataset.

Large thermal inequities existed for all of the urban regions we investigated. Low-income and Latinx populations were the most impacted (Figure 3 and Figure 4). On average, we found the poorest 10% of neighborhoods in each region to be about 2.2 °C (4 °F) hotter than the wealthiest on both extreme heat days and average summer days. The difference was as high as 3.3 to 4 °C (6–7 °F) (Figure 3 and Figure 4) in California metro areas such as Palm Springs and the Inland Empire. The neighborhoods that had the smallest differences were seen in Albuquerque and El Paso, which also were the cities with the highest Latinx populations at 47.6% for Albuquerque and 83.6% for El Paso.

The top deciles of Latinx neighborhoods in each region were just under 2.2 °C (4 °F) hotter on average than the lowest decile in Latinx population, with disparities of up to nearly 7 °F in some cases. Low-income and high-Latinx population neighborhoods were substantially correlated (−0.4), with highly Latinx neighborhoods being among the poorest for many cities throughout the Southwestern U.S.

Top-decile neighborhoods in the Black population saw modestly warmer temperatures of 0.5 to 1.1 °C (1–2 °F) compared to neighborhoods with the lowest Black populations. However, Black populations throughout the southwestern metropolitan areas are relatively small, and even the top-decile neighborhoods still had low percentages of Black population. Salt Lake City (<2.5% even for the top decile), Albuquerque (<3%), and El Paso (<3%) are good examples of this.

Temperature disparities were far less at nighttime. On average, across these 20 Southwestern U.S. metro areas, we found the poorest 10% of neighborhoods in each region to be 2 °F hotter than the wealthiest 10% at night, although this difference was as high as 2.5 °C (4 °F) in Los Angeles, the Inland Empire, and Palm Springs. We found the top decile of Latinx neighborhoods in each region to be 1 °C (1.8 °F) warmer on average than the lowest decile at night, although up to nearly 4.5 °F in some urban regions. At night, there did not appear to be a substantial temperature difference between the most heavily Black neighborhoods and the least (<0.3 °F), again with the caveat that Black populations in most of these metro areas are relatively small.

All of our models and statistics support these thermal equity observations. We used regression models to elaborate on the benchmark set by the comparison of means test. These models are able to reveal clues about the direct associations between certain demographics and heat (see Table A1 in the Appendix A). Across our sample of Southwestern U.S. metropolitan areas, we found that a 10% decrease in income is associated with a 0.22 degrees Fahrenheit increase in temperature during daytime for both extreme heat and average heat days. At night, temperatures were 0.09 degrees Fahrenheit warmer for every 10% decrease in neighborhood income. Regression models controlling for spatial dependency yielded similar findings. We found negative associations between temperature and income across all three temperature scenarios, meaning that higher neighborhood income results in cooler temperatures.

El Paso, Texas, on an average heat day and an extreme heat day had the highest regression coefficients for Latinx neighborhoods, indicating a large thermal inequity for Latinx neighborhoods. In both temperature scenarios, Phoenix had the second highest regression coefficients for Latinx populations outside of California. Denver, Colorado, had the highest income regression values for both average and extreme heat (Figure 5 and Figure 6) indicating a high thermal gradient between wealthier and poorer areas. Reno, Salt Lake City, and Albuquerque all also had high regression coefficients as well, indicating wealthier neighborhoods were substantially cooler than lower-income neighborhoods. All four of these cities configure with wealthier neighborhoods at higher elevations as the metro area encroaches on the foothills of their local mountain ranges.

Since low-income and Latinx neighborhoods in many parts of the Southwest have little green space, high amounts of impervious surfaces, and higher urban density (e.g., [18,25], it seems likely that these factors explain much about thermal disparities. Satellite imagery could verify this assumption, and further research could develop such an analysis. The implication would be that programs to increase vegetation within disadvantaged neighborhoods and reduce or lighten pavements and rooftops could help reduce thermal disparities between neighborhoods of different socioeconomic characteristics.

### California

Because California represents a large proportion of our sample and is distinct from the rest of the Southwest in terms of its climate and demographics, we examined a subset of the data representing California cities and compared it with data for the rest of the Southwest. We ran additional regression models that separated California neighborhoods from the rest of the sample. We found that California’s urban regions had much larger temperature differences between the wealthiest and poorest neighborhoods than regions in the rest of the Southwest. For extreme heat days, the poorest decile of California neighborhoods in each region were 2.6 °C (4.7 °F) hotter on average than the wealthiest decile neighborhoods, compared to a mean difference of 1.7 °C (3.1 °F) in other Southwestern urban areas. For average heat days, the mean difference was 2.5 °C (4.5 °F) in California metropolitan areas, compared with 2.1 °C (3.8 °F) in other southwestern cities. Nighttime thermal differences in California were 1.3 °C (2.2 °F) on average, compared to 1 °C (1.8 °F) for Southwestern urban regions outside the state.

Temperature disparities experienced by Latinx California neighborhoods were similarly larger than those experienced outside the state. The highest-decile block groups in terms of Latinx population in the California sample were 2.6 and 2.3 °C (4.7° and 4.1 °F) warmer on extreme heat and average heat days respectively, compared with 1.5 and 1.3 °C (2.7 and 3.1 °F) for Southwestern cities outside of California. The largest differences occurred in the Inland Empire (3.6 °C (6.5 °F for an average heat day)), Palm Springs (3.8 °C (6.9 °F for an extreme heat day)), and Los Angeles (3.6 °C (6.5 °F for an extreme heat day)). Nighttime temperature disparities were not as great: 1.1 °C (2 °F) for California Latinx neighborhoods compared to 0.6 °C (1.1 °F) for Southwestern Latinx neighborhoods outside the state.

Black neighborhoods in California’s urban regions experienced only slightly greater temperature disparities than in other Southwestern locations. On average heat days, California’s top-decile Black neighborhoods were 0.9 °C (1.6 °F) warmer than their lowest-decile counterparts, while the average disparity in other Southwestern metro areas was 0.5 °C (0.8 °F). On extreme heat days, both groups’ disparities were just under 1.1 °C (2 °F).

Regression analysis showed that for Latinx populations the associations for California neighborhoods were more than twice as large (6.41 vs. 2.92) as those for non-California neighborhoods on extreme heat days, and nearly three times as large (3.03 vs. 1.084) for nighttime temperatures. Our model predicts heavily Latinx neighborhoods (75th percentile) in California to be 1.8 °C (3.2 °F) degrees warmer than less densely populated Latinx neighborhoods (25th percentile) on an extreme heat day. This is nearly three times the disparity for Latinx neighborhoods in Southwestern urban areas outside of California. On average heat days, our model predicts that 75th percentile Latinx neighborhoods will be 1.7 °C (3.1 °F) degrees warmer than less densely Latinx neighborhoods in California, compared with a similar disparity of 0.7 °C (1.3 °F) degrees within metro regions in the rest of the Southwest. For Black populations under extreme heat and average heat days, the California coefficients were double those of non-California cities (see Appendix A).

For average heat days, Palm Springs, Bakersfield, and Fresno had the highest spatial regression coefficients, substantially higher than other California cities (Figure 7). These same cities also had substantially higher coefficients in the extreme heat scenario indicating that there were greater thermal inequities in these three metro areas. In both daytime scenarios, Sacramento had the lowest regression coefficients. Palm Springs and the Inland Empire had the highest regression coefficients for average and extreme heat days (Figure 5 and Figure 6), indicating that the highest income-based thermal inequities existed in these metro regions.

Reasons for greater thermal inequities in California likely concern vegetation and geography. While irrigated green space is not uncommon in Las Vegas and Phoenix, high-income California neighborhoods typically have more vegetation than their counterparts in other Southwestern urban regions. A history of state- and federally funded water projects in California have made cheap water available to many residential developments. Additionally, California’s physical environment appears to play a role in that higher-income neighborhoods are often located close to parks, water bodies, natural green space, or irrigated farmland. Urban regions such as Bakersfield and Fresno surround extensive irrigated agriculture. For urban regions such as San Jose and Palm Springs, nearby mountains may ameliorate temperatures within high-income neighborhoods. Wealthy neighborhoods in Sacramento may be disproportionately cooled by the American and Sacramento river greenways. In Los Angeles and San Diego, higher-income neighborhoods are near the Pacific Ocean. Further research will be necessary to more precisely identify the role of such features in creating greater thermal inequities within California’s urban areas (Figure 6 and Figure 8).

The regression model run uses sociodemographic variables from the ACS dataset.

## 4. Conclusions

Our in-depth analysis identifies large and consistent variations often 2.2–3.9 °C (4–7 °F) between neighborhoods of different class and racial composition within Southwestern U.S. urban regions. Although previous literature has documented thermal inequities elsewhere, we examined the largest number of urban regions at the most detailed scale to date under both day- and nighttime temperature scenarios. On both average and extreme heat days, low-income neighborhoods of the Southwestern U.S. urban regions experience temperatures that are 3.9 °C (7 °F) higher in some cases and nearly 2.2 °C (4 °F) higher on average than those experienced by wealthy neighborhoods. The pattern is similar for neighborhoods with different levels of Latinx population. Income has a statistically significant relationship with average daytime heat, extreme heat, and average nighttime heat. This trend is present in 56 of the 59 spatial error models that we ran. The same significance holds in regards to Latinx population. We found thermal inequalities to be most stark in California, perhaps because of greater amounts of irrigated landscape and agriculture as well as geographical factors mentioned earlier. Our in-depth analysis highlights the extent of these disparities under summer daytime, nighttime, and extreme heat scenarios within 20 dryland urban regions at particular risk of high heat. We also looked specifically at percent Latinx, a marginalized group throughout the Southwestern United States. We also utilized a spatial error model to account for spatial autocorrelation, which was prevalent in our biophysical and sociodemographic variables. Finally, although we presented results in majority dryland cities, we also included the humid cities of Houston and Dallas.

The inequalities are most stark in California, perhaps because of greater amounts of irrigated landscape and agriculture as well as geographical factors, as mentioned earlier. California cities also show huge differences between the top decile and bottom decile of Latinx population neighborhoods compared to the rest of the Southwest. These two variables appear highly correlated, as the highest-Latinx neighborhoods are also the poorest for many cities throughout the Southwest.

Our study contributes strong new evidence to the body of literature on climate justice. Certain neighborhoods are shouldering a disproportionate burden of urban heat risk [25] This disproportionate risk exists regardless of urban heat islands, i.e., whether the city is hotter than surrounding rural landscapes. Policy makers and stakeholders may utilize these findings for targeted mitigation protocols, for example, involving increased vegetation, shading of paved surfaces, use of light-colored urban surface materials, and built form that creates ground-level shade. Cooling centers and other public health interventions may also be needed. Policies set forth to allocate resources and funds to neighborhoods that meet the criteria of certain income thresholds or demographics could be effective in smoothing the thermal inequity that is present in many cities.

While our study covers a robust set of metropolitan areas over a large geographic region, other cities within the U.S. or elsewhere may show somewhat different patterns. More research is needed to fully develop an understanding of neighborhood-scale thermal disparities within urban regions globally. Additionally, the satellite imagery used here, as in the large majority of urban heat studies, measures surface temperatures rather than ambient air temperatures. The latter is most relevant to human comfort and health, but must typically be measured in situ or estimated through modeling. Such techniques are difficult over large metropolitan areas. Finally, the demographics explored here are only a small subsection of the demographics that are potentially at risk for unequal exposure to heat. Other variables of potential interest to future investigators include education, age, disability, citizenship, median housing value, and percentage of single-parent households.

## Figures and Tables

**Figure 1 ijerph-18-00941-f001:**
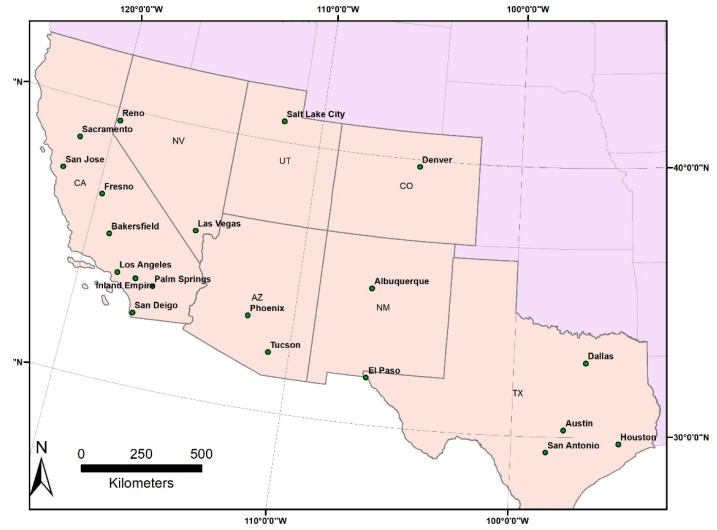
The 20 Southwestern cities under investigation.

**Figure 2 ijerph-18-00941-f002:**
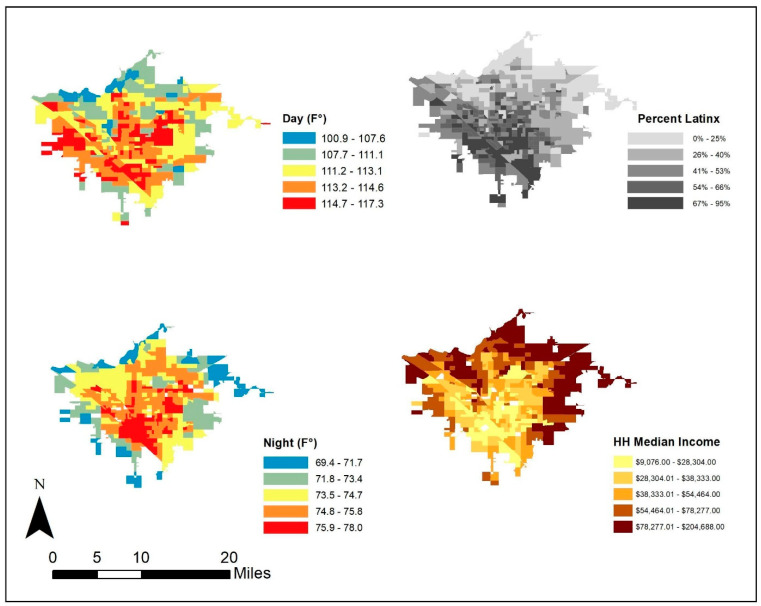
Fresno, California, located at 36.7° N and 119.8° W: surface temperature for an extreme heat day.

**Figure 3 ijerph-18-00941-f003:**
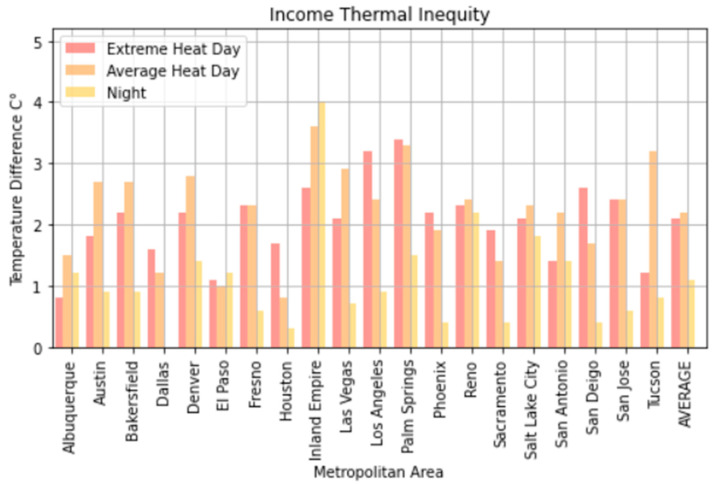
Income thermal inequity—a comparison of the temperature difference between the top 10 percentile of block groups and bottom 10 percentiles of block group based on income.

**Figure 4 ijerph-18-00941-f004:**
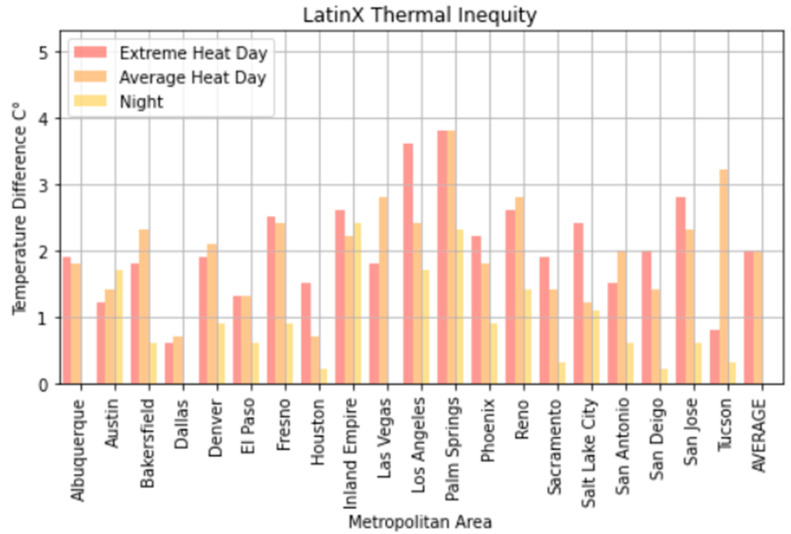
Latinx thermal inequity—a comparison of the temperature difference between the top 10 percentile of block groups and bottom 10 percentiles of Latinx block group population.

**Figure 5 ijerph-18-00941-f005:**
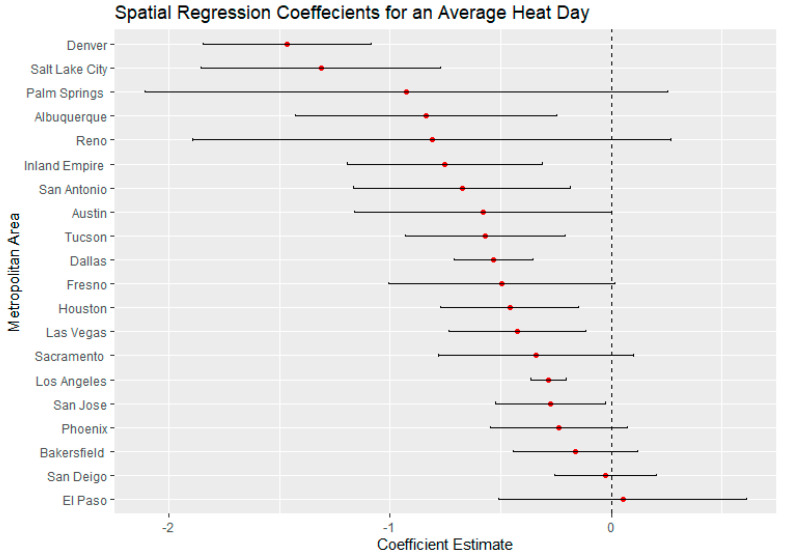
Spatial regression coefficients for median income during an average heat day in 20 Southwest metropolitan areas.

**Figure 6 ijerph-18-00941-f006:**
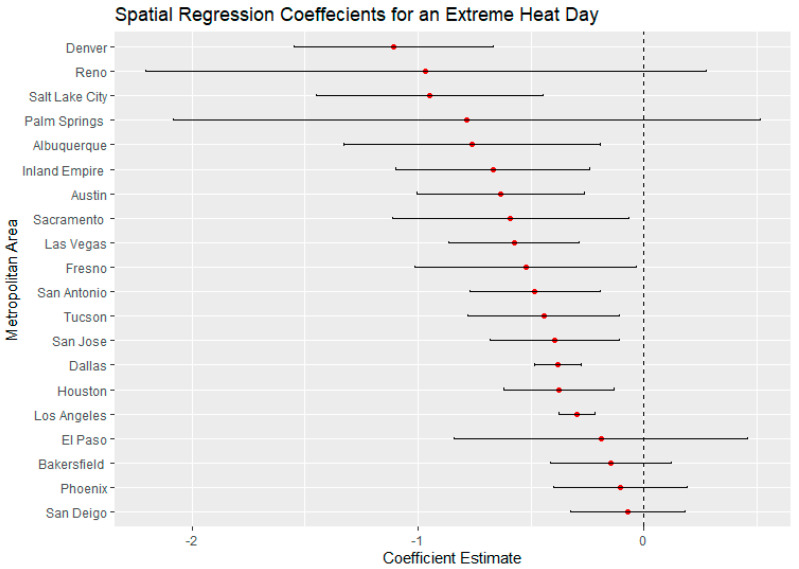
Spatial regression coefficients for median income during an extreme heat day in 20 Southwest metropolitan areas.

**Figure 7 ijerph-18-00941-f007:**
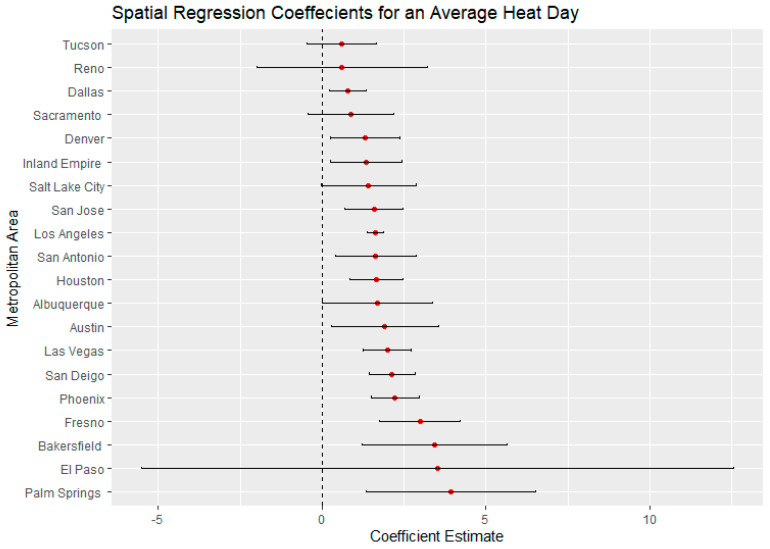
Spatial regression coefficients for Latinx populations during an average heat day in 20 Southwest metropolitan areas.

**Figure 8 ijerph-18-00941-f008:**
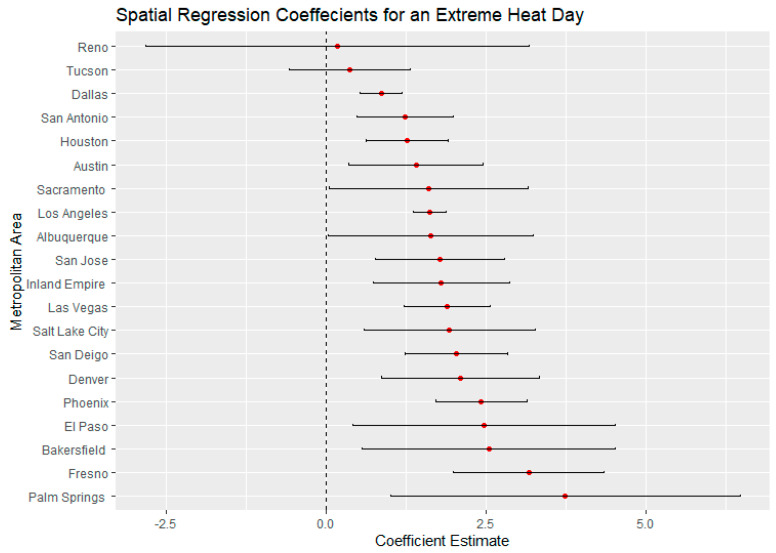
Spatial regression coefficients for Latinx populations during an extreme heat day in 20 Southwest metropolitan areas.

## Data Availability

Data available here: https://drive.google.com/drive/folders/1gIQRHDWV1TNArFS5R8GNGtV_p69GAmLv?usp=sharing.

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
