# Peer review of "Dimensions of Thermal Inequity: Neighborhood Social Demographics and Urban Heat in the Southwestern U.S."

_ijerph, 2021, doi:10.3390/ijerph18030941_

Round 1
Reviewer 1 Report
This paper presents a study on the thermal inequity in 20 southwestern U.S. metropolitan regions using remote sensing. I have reviewed this manuscript before. Compared with the previous one, this resubmitted version made some improvement but was not substantially improved. I do not think this manuscript can be considered for publication because the major problems are still unsolved. The method and analysis are still rough. It also suffers from a poor writing style. - The structure of the Introduction Section is a little confusing. The authors need to reorganize this section. - There are still a lot of writing and format mistakes. - Section 2.2: Both Equation (1) and (2) are wrong. - Section 2.3: The methods are not clearly illustrated. - The resolution of Figure 3 and 4 are too low. - Results and Discussion Section. The section mostly descries the result. In-depth discussion should be given. Especially on the different performance of different cities. Moreover, the novelty should be justified by comparing the proposed method and results with other studies. The limitations of this study should be given. In addition, the thermal inequity results are discussed with the vegetation cover. The simple analysis is not enough. The authors shall calculate vegetation information (NDVI or vegetation cover) from the same Landsat images and quantitatively analyze the relationship with thermal inequity.Author Response
Reviewer 1
Comment
Methods and analysis are still rough. It also suffers from a poor writing style.
Response
This reviewer’s comments are contrary to those of the other two reviewers, who believed both method and writing were acceptable. For example, reviewer 2 stated “the manuscript is well written and the results are presented clearly - the article is ready for publication.” Reviewer 3 stated “The authors responded to my notes, so I believe the text is ready for acceptance”
Comment
Both Equation (1) and (2) are wrong
Response
These equations are correct as supported by our references which have several hundred citations themselves in peer reviewed journals. For example Li et al. 2016 used this method for urban heat within shanghai.
Additionally Zhou et al. 2011used this for the region of sacramento. Lastly Bokaie et al. 2016 used this method for looking at LST in Tehran, Iran.
Li, J., Song, C., Cao, L., Zhu, F., Meng, X., & Wu, J. (2011). Impacts of landscape structure on surface urban heat islands: A case study of Shanghai, China. Remote sensing of environment, 115(12), 3249-3263.
Zhou, W., Wang, J., & Cadenasso, M. L. (2017). Effects of the spatial configuration of trees on urban heat mitigation: A comparative study. Remote Sensing of Environment, 195, 1-12.
Bokaie, M., Zarkesh, M. K., Arasteh, P. D., & Hosseini, A. (2016). Assessment of urban heat island based on the relationship between land surface temperature and land use/land cover in Tehran. Sustainable Cities and Society, 23, 94-104.
Comment
Section 2.3: The methods are not clearly illustrated. - The resolution of Figure 3 and 4 are too low.
Response
We feel that the methods are clearly explained. For example lines 150-209 we explain acquisition, processing, and analysis concisely. Reviewer 2 stated “ All comments regarding the original version of the manuscript have been taken into account by the authors. The description of the methodology, results & discussion and conclusion has been extended.”
The resolution of the figures are 300 DPI and clearly show and illustrate the metrics.
Comment
The section mostly describes the result. In-depth discussion should be given.
Response
The results and discussions section entail a significant amount of discussion, and includes comparison with other studies. Lines 361 to 381 specifically highlight this.
Comment
The limitations of this study should be given. In addition, the thermal inequity results are discussed with the vegetation cover. The simple analysis is not enough. The authors shall calculate vegetation information (NDVI or vegetation cover) from the same Landsat images and quantitatively analyze the relationship with thermal inequity.
Response
Limitations of this study are discussed on lines 391-410. Adding NDVI we feel will be a publication in itself and require additional processing time. Since this focuses on the interactions of temperature and social variables and not biophysical variables such as NDVI we feel that this is a substantial addition to the literature

Reviewer 2 Report
All comments regarding the original version of the manuscript have been taken into account by the authors. The description of the methodology, results & discussion and conclusion has been extended. In my opinion, the manuscript is well written and the results are presented clearly - the article is ready for publication.
Author Response
Thank you for the kind words and helpful suggestions. We really appreciate you reviewing our article.
Reviewer 3 Report
The authors responded to my notes, so I believe the text is ready for acceptance.
Author Response
Thank you so much for all of your helpful edits.
This manuscript is a resubmission of an earlier submission. The following is a list of the peer review reports and author responses from that submission.
Round 1
Reviewer 1 Report
This paper presents a study on the thermal inequity in 20 southwestern U.S. metropolitan regions using remote sensing. It's an very interesting study but the writing and Analysis are rough. It is not appropriate for publication until substantial improvements are made.
- Surface urban heat island (SUHI) indicated by remotely sensed land surface temperature are not obvious in some arid cities. For cities in humid areas, urban areas are characterized by impervious surfaces and surrounded by vegetation. Impervious surfaces generally show high daytime surface temperature while vegetation’s show low temperature, resulting in obviously SUHI. However, for arid cities, the vegetation cover in urban areas may be higher than in suburb areas and therefore the SUHI are not obvious. In the Introduction Section, I suggest to delete the UHI contents and indicate the relationship between green space and temperature.
- The structure of the Introduction Section is a little confusing. The authors need to reorganize this section.
- There are some tiny mistakes. eg, L63 (Harlan et al. 2007) found older ...; L120, ... of the 20 metropolitan regions. selecting the warmest ...
- Section 2.2. The headline should be changed to "Apparent brightness temperature calculation" or "At-sensor brightness temperature calculation". In this study, only apparent brightness temperature was calculated, which did not used the single-channel method for land surface temperature retrieval. Apparent brightness temperature is not an idea indicator for thermal environment because it does not eliminate the influent of atmospheric absorption, atmospheric radiance and also the land surface emissivity. If possible, I suggest the authors to retrieve land surface temperature to indicate thermal environment.
- Equation 1, the variables in this equation need to be explained.
- Section 2.3. The headline should be changed to "Thermal inequity analysis". This section needs to be rewritten to clear illustrate the method.
- In the daytime, impervious surface and soil generally show higher temperature than vegetation but at night the temperature difference is much smaller. Even without this study, people can infer that nighttime temperature difference is small and therefore thermal inequity is small. I suggest to remove the analysis on nighttime heat.
- Results and Discussion Section. The section mostly descries the result. In-depth discussion should be given. Especially on the different performance of different cities. Moreover, the novelty should be justified by comparing the proposed method and results with other studies. The limitations of this study should be given.
- Results and Discussion Section. The thermal inequity results are discussed with the vegetation cover. Why not directly derive vegetation information (NDVI or vegetation cover) from the same Landsat images and quantitatively analyze its relationship with thermal inequity?
- Figure 3. It's hard to read due to the low resolution.
Author Response
Surface urban heat island (SUHI) indicated by remotely sensed land surface temperature are not obvious in some arid cities. For cities in humid areas, urban areas are characterized by impervious surfaces and surrounded by vegetation. Impervious surfaces generally show high daytime surface temperature while vegetation’s show low temperature, resulting in obviously SUHI. However, for arid cities, the vegetation cover in urban areas may be higher than in suburb areas and therefore the SUHI are not obvious. In the Introduction Section, I suggest to delete the UHI contents and indicate the relationship between green space and temperature
Response: Yes, we agree that SUHIs for dryland cities have very different characteristics, and have done previous research on this. We have edited the paper to clarify that heat is what really matters in terms of human comfort and health. However, the public is familiar with the UHI phenomenon, and this is what makes certain urban neighborhoods are hotter than others, so we feel this is still an essential concept to the paper.
The structure of the Introduction Section is a little confusing. The authors need to reorganize this section.
Response: We have added some clarifying materials and made additional edits, but feel the organization is fundamentally sound.
There are some tiny mistakes. eg, L63 (Harlan et al. 2007) found older ...; L120, ... of the 20 metropolitan regions. selecting the warmest ..
Response: Thank you for pointing these out we have fixed these small errors at lines 63 and 120 and believe they are now more clear and well suited.
- Section 2.2. The headline should be changed to "Apparent brightness temperature calculation" or "At-sensor brightness temperature calculation". In this study, only apparent brightness temperature was calculated, which did not used the single-channel method for land surface temperature retrieval. Apparent brightness temperature is not an idea indicator for thermal environment because it does not eliminate the influent of atmospheric absorption, atmospheric radiance and also the land surface emissivity. If possible, I suggest the authors to retrieve land surface temperature to indicate thermal environment.
Response: This is a great comment. Thank you for catching. In an effort to reduce text we took the conversion equation to LST be put but have but that back in. Please keep in mind that results will not change due to us already using LST.
Equation 1, the variables in this equation need to be explained
Response: Variable explanation added in the text K1 and k2 are Band-specific thermal conversion constants and variables.
Section 2.3. The headline should be changed to "Thermal inequity analysis". This section needs to be rewritten to clear illustrate the method.
Response: Thank you for the comment. We have restructured this section as well as changing the heading to the suggested title.
In the daytime, impervious surface and soil generally show higher temperature than vegetation but at night the temperature difference is much smaller. Even without this study, people can infer that nighttime temperature difference is small and therefore thermal inequity is small. I suggest to remove the analysis on nighttime heat.
Response: Thank you very much for the comment. Our analysis shows that the nighttime range of temperatures can still be 6-7 degrees F across metropolitan areas, so we feel it’s important to keep this in. See Figure 2 for example.
Results and Discussion Section. The section mostly descries the result. In-depth discussion should be given. Especially on the different performance of different cities. Moreover, the novelty should be justified by comparing the proposed method and results with other studies. The limitations of this study should be given.
Response: Thank you for this excellent suggestion, in the text we have compared cities as well as variables and included limitations in the text.
Results and Discussion Section. The thermal inequity results are discussed with the vegetation cover. Why not directly derive vegetation information (NDVI or vegetation cover) from the same Landsat images and quantitatively analyze its relationship with thermal inequity?
Response: That is an excellent suggestion, but goes beyond the scope of this paper, and could easily amount to a whole paper in itself. We will consider adding that as a further component of this research.
Figure 3. It's hard to read due to the low resolution.
Response: Figure 3 has been enhanced for readability and is now broken down to figure 3 and figure 4.
Reviewer 2 Report
The manuscript deals with the issue of thermal inequity, which is especially important in regions with high temperatures. The research results clearly show that the socio-demographic and economic characteristics are correlated with thermal inequitiy. The manuscript is interesting but should be improved.
- In the Introduction chapter, the authors use the Celsius scale in relation to temperatures, while in their research they use the Fahrenheit scale. Wherever degrees Celsius is given, the value in the Fahrenheit scale should be given in brackets.
- Abstract - the authors write about three temperature scenarios (average summer heat, extreme summer heat, and average nighttime heat). Does average nighttime heat also apply to summer? Given the text on line 102, yes. This should be clearly described.
- Whether the night-time temperature test was carried out on average hot days or on days with extreme heat - this should be clarified. This is important because the temperature during the day can translate into the temperature at night.
- The authors should provide the temperature ranges for the 20 metropolitan regions - average summer heat, extreme summer heat, and average nighttime heat. This could be Table 2 in Appendices. This is vital information. The term "an average summer day" is very imprecise, for example.
- Figure 3 - mistake in signature. Instead of right it should be top, and left - bottom. Additionally, the titles placed directly above the charts should be the same as in the figure caption.
- In the Introduction chapter, authors should describe the actions that are taken to reduce thermal inequities - based on the literature analysis.
Author Response
In the Introduction chapter, the authors use the Celsius scale in relation to temperatures, while in their research they use the Fahrenheit scale. Wherever degrees Celsius is given, the value in the Fahrenheit scale should be given in brackets.
Response: This is a great comment. We have now provided temperatures in both scales throughout the text.
Abstract - the authors write about three temperature scenarios (average summer heat, extreme summer heat, and average nighttime heat). Does average nighttime heat also apply to summer? Given the text on line 102, yes. This should be clearly described.
Response: Thank you for catching this. We have added this explanation within the abstract, and now feel that it is clearly described.
Whether the night-time temperature test was carried out on average hot days or on days with extreme heat - this should be clarified. This is important because the temperature during the day can translate into the temperature at night.
Response: Thank you for the comment. We agree that is is very important and thus has been clearly articulated throughout the text.
The authors should provide the temperature ranges for the 20 metropolitan regions - average summer heat, extreme summer heat, and average nighttime heat. This could be Table 2 in Appendices. This is vital information. The term "an average summer day" is very imprecise, for example.
Response: Thank you for the suggestion. we have added both the Celsius and Fahrenheit ranges in Table 2 located in the Appendix.
Figure 3 - mistake in signature. Instead of right it should be top, and left - bottom. Additionally, the titles placed directly above the charts should be the same as in the figure caption.
Response: Figure 3 has been adjusted to include the recommendations. The figure caption has also been enhanced to aid in more clarity.
In the Introduction chapter, authors should describe the actions that are taken to reduce thermal inequities - based on the literature analysis.
Response: Thank you for this insight. We have made additions to the introduction section to reflect these recommendations.
Reviewer 3 Report
The article has scientific merit and the issue has growing importance in the climate literature. For this I congratulate the authors. Requires substantial corrections.
On lines 41 and 42, the authors forget that the urban heat island is not an exclusive phenomenon of the temperate environment. This needs to be reviewed because is a conceptual confusion.
I disagree that the data is presented in °F, especially because it was based on works that used °C. I think conversion is appropriate.
I believe there is a problem with the formula used. You need to review it.
Maps have problems like the lack of latitude and longitude grid.
Figure 4 has image quality problems. They need to improve the resolution and increase the size of words.
The concept of risk appears at the end of the text and needs a reference (line 289).
The analysis is the treatment of the data is adequate enough.
I chose to make these and other indications in the text.

Author Response
On lines 41 and 42, the authors forget that the urban heat island is not an exclusive phenomenon of the temperate environment. This needs to be reviewed because is a conceptual confusion.
Response: Thank you for the comment. This has been adjusted in the text to address the conceptual confusion
I disagree that the data is presented in °F, especially because it was based on works that used °C. I think conversion is appropriate.
Response: The authors have converted and provide all temperature readings in celsius including the new table 2 for temperature ranges in the appendix
I believe there is a problem with the formula used. You need to review it.
Response: thank you for catching, Formula has been revised in text to show that we used land surface temperature. Please keep in mind we used LST but left out the second step in the equation mistakenly in text. Also the appropriate citation has been added
Maps have problems like the lack of latitude and longitude grid.
Response: Lat and long grids have been added to map 1 but the authors feel that map 2 being only a single metro area does not require lat and long. The authors have added the latitude and longitude location in image caption
Figure 4 has image quality problems. They need to improve the resolution and increase the size of words.
Response: Resolution has been rectified and broken into seperate larger images for readability
The concept of risk appears at the end of the text and needs a reference (line 289).
Response: We have added the appropriate reference
I understand based on the references you have presented, but the urban heat island is not an exclusive phenomenon of the temperate environment. Tropical cities also register the phenomenon, with numerous studies proving its persistence and great magnitudes.
Response: Excellent point and a worthwhile addition to the paper. we have added this acknowledgement in the introduction
Note that the studies cited presented results in degrees Celsius
Response: We have converted to Celsius for analysis to match
Reviewer 4 Report
A very interesting approach to environmental justice and highlighting inequalities by combining satellite data and census data.
The methodology is sound and providing details of every step as well as assumptions and limitations of the research.
An area for improvement is related to the regression models discussed, which could be presented and the correlation in the model.
Figure 3 should be enhanced with more details, such as what is the comparison in the graph? The Y axis should read temperature difference.
I would suggest to increase the size of Figure 4 for readability.
The conclusions are supported by the results and point out the highlights of the research
Author Response
A very interesting approach to environmental justice and highlighting inequalities by combining satellite data and census data.
Response: Thank you! we appreciate it.
The methodology is sound and providing details of every step as well as assumptions and limitations of the research.
Response: Thank you again for the compliment
An area for improvement is related to the regression models discussed, which could be presented and the correlation in the model.
Response: We have presented the regression model in the text as well as adding R-squraed values in table 1 in the appendix
Figure 3 should be enhanced with more details, such as what is the comparison in the graph? The Y axis should read temperature difference.
Response: We have adjusted the graph axis and have written a more robust image caption to articulate the comparison better.
I would suggest to increase the size of Figure 4 for readability.
Response: Thank you for the suggestion, the figure size has been enhanced
The conclusions are supported by the results and point out the highlights of the research
Response: Once again, thank you very much!